# Profile of Hospital Admissions Due to Preterm Labor and Delivery in England

**DOI:** 10.3390/healthcare11020163

**Published:** 2023-01-05

**Authors:** Abdallah Y. Naser, Hassan Al-Shehri, Noora Altamimi, Anas Alrasheed, Lama Albalawi

**Affiliations:** 1Department of Applied Pharmaceutical Sciences and Clinical Pharmacy, Faculty of Pharmacy, Isra University, Amman 11622, Jordan; 2Department of Pediatrics, College of Medicine, Imam Mohammad Ibn Saud Islamic University, Riyadh 11564, Saudi Arabia; 3College of Medicine, Imam Mohammad Ibn Saud Islamic University, Riyadh 11564, Saudi Arabia

**Keywords:** admissions, hospital, preterm, labor, delivery, England

## Abstract

Objectives: Preterm labor and delivery are associated with various short- and long-term complications in neonates and infants. This research aimed to look at the trends in preterm labor and birth-related hospitalizations in England. Material and Methods: The Hospital Episode Statistics database was used to extract hospital admission data for the ecological study of preterm labor and delivery between April 2012 and April 2020 in England. Results: The overall admission rates decreased by 26.2%, from 14,210 in 2012 (CI: 99.18–102.49) to 10,490 in 2020 (CI: 73.02–75.87) per 100,000 individuals. Hospitalizations were frequently caused by spontaneous labor with preterm delivery, spontaneous labor without delivery, and preterm delivery without spontaneous labor (68.9%, 20.6%, and 9.6%, respectively). The rate of hospital admission due to preterm delivery without spontaneous labor, preterm labor without delivery, preterm spontaneous labor with preterm delivery, and preterm spontaneous labor with term delivery decreased by 32.0%, 26.9%, 24.4%, and 14.7%, respectively. Women aged 25–29 years accounted for most hospital admissions. Conclusion: Preterm labor and delivery-related hospital admissions rates have significantly decreased over the past decade. Women in the reproductive age range of 25–34 years were more prone to hospital admission followed by preterm labor due to various reasons.

## 1. Introduction

Preterm labor is described as regular contractions resulting from dilating or curving of the cervix between 20–37 weeks of gestation [1,2,3]. It is classified as late or early preterm labor based on the time of its occurrence. Late preterm delivery is defined as when birth occurs between 34–36 weeks, whereas early premature delivery occurs when the baby is born before 33 weeks [4]. Preterm labor and delivery are associated with various complications. Neonatal complications include congenital anomalies, bronchopulmonary dysplasia, intraventricular hemorrhage, weak growth, bronchopulmonary dysplasia, necrotizing enterocolitis, and retinopathy of prematurity [5]. Infant complications include behavioral issues and an impaired neurodevelopmental outcome, while maternal complications results in an higher risk of cardiovascular morbidity and mortality [4]. In developed countries, preterm labor reportedly occurs in 5–9% of gestations [6]. About 75–80% of all significant neonatal morbidity and neonatal deaths are due to preterm labor [6,7]. Moreover, preterm labor is a leading cause of antenatal hospitalization among pregnant women, which puts considerable economic and social burdens on the community [8].

Preterm labor leads to preterm birth. Thus, its proper diagnosis and effective management are crucial [8]. Data extracted from the Office of National Statistics showed that in 2014, preterm delivery occurred in 8.7% of 52,249 births in England [9]. In 2010, about 15 million premature births were reported, and about 1 million of them died due to ensuing complications [10]. In 2019, there were 940,000 deaths of children younger than five years due to complications followed by preterm birth [11]. Many preterm children live with lifelong disabilities, including hearing problems, visual problems, and learning disabilities. In the United Kingdom (UK), one out of every 12 babies has a preterm birth [12]. Premature babies usually require admission to an intensive care unit as they can have severe health problems, including respiratory problems, necrotizing colitis, and sepsis [4].

Apart from being a debilitating obstetric problem, preterm births put a severe financial and emotional burden on the family [13]. The sequelae of preterm births have substantial long-term economic consequences for the healthcare organization and for society in general [14]. The adaptation of standardized protocols for evaluating preterm labor can reduce hospitalization rates and result in a significant reduction in expenses [15]. In addition, women with premature deliveries are at risk of unhealthy physical and emotional outcomes, leading to the scarce use of postnatal services and support [16]. As a result, postnatal depression is quite common in women with preterm deliveries. The determination of preterm births is challenging in most scenarios where ultrasound data is not readily available throughout the pregnancy. An estimation of gestational age may be inaccurate where pregnancies are affected by conditions such as intrauterine growth restrictions [17]. Global collaborative efforts are underway to define gestation age more specifically and focus on areas where preterm births are common.

Preterm birth is a crucial global problem due to its high morbidity and mortality and the ensuing socioeconomic burden. Though tangible metrics, such as neonatal and childhood mortality rates due to premature birth, provide a substantial measure of this burden, the associated psychological and financial aspects cannot be ignored. Continuous research and publication of new data on preterm labor and births ensures the attention of the world community to prevent and manage such cases efficiently [18]. Previous studies in the United Kingdom have examined the hospitalization profile among different disease areas, however, none of them examined admissions related to preterm labor and delivery [19,20,21]. In this regard, the current study aims to generate an admission profile of preterm labor and delivery in England between 2012 and 2020, and analyze the general trend of such cases with respect to demographic aspects of the mothers.

## 2. Materials and Methods

### 2.1. Study Source and Population Demography

This was an ecological study that used publicly available data in the UK. Herein, data was taken from the Hospital Episode Statistics (HES) in England [9] between April 2012 and April 2020. The HES reports all hospitalization episodes in England and makes them freely available to the public for the purpose of healthcare analysis [9]. The database comprised hospital admission data for premature labor and delivery patients of all ages, which are grouped into various segments. Data is reported for females categorized according to their ages as: 10–14 years, 15 years, 16 years, 17 years, 18 years, 19 years, 20–24 years, 25–29 years, 30–34 years, 35–39 years, 40–44 years, and 45–57 years. The 10th revision of the International Statistical Classification of Diseases and Related Health Problems (ICD-10) 5th edition (National Health Service (NHS) was used to identify preterm labor and delivery-related hospital admissions (O60.0–O60.3) [22]. The HES database records all hospital admissions performed by NHS trusts and NHS-funded independent sectors. Since 1999/2000, information regarding hospital admissions in England is accessible. The data includes demographics, diagnosis, procedures, and length of stay. The HES data are routinely validated and checked for precision [9]. After obtaining mid-year population data from the Office for National Statistics (ONS) for females aged 10–49 years between 2012 and 2020, we calculated the annual hospital admission rate for preterm labor and delivery [23].

### 2.2. Statistical Analysis

The number of admissions for each age group were divided by the mid-year population of that group for the respective year to calculate hospital admission rates with 95% confidence intervals (CIs). Between 2012 and 2020, the hospital admission rates, other admission methods, emergency, FCE bed days, waiting list, and scheduled admission were compared using the chi-squared test. All analyses used SPSS version 27 (IBM Corp., Armonk, NY, USA).

## 3. Results

During the study period, there were 97,720 admissions related to preterm labor and delivery in England, with an average of 12,215 per year. In 2020, the total annual number of preterm labor and delivery hospital admissions for different etiologies decreased by 26.2% from 14,210 in 2012 to 10,490 in 2020, showing a decrease in the rate of hospital admission [from 100.84 (CI 99.18–102.49) in 2012 to 74.44 (CI 73.02–75.87) in 2020 per 100,000 individuals (*p ≤* 0.05)].

Common preterm labor and delivery hospital admissions reasons were preterm spontaneous labor with preterm delivery, preterm labor without delivery, and preterm delivery without spontaneous labor, which accounted for 68.9%, 20.6%, and 9.6%, respectively (Table 1).

Throughout the study period, the preterm labor and delivery hospital admissions rate decreased as follows: preterm delivery without spontaneous labor by 32.0%, preterm labor without delivery by 26.9%, preterm spontaneous labor with preterm delivery by 25.4%, and preterm spontaneous labor with term delivery by 14.7% (Figure 1).

Regarding age group differences for preterm labor and delivery hospital admissions, the age group 25–29 accounted for 28.4% of the total number of preterm labor and delivery hospital admissions, followed by the age group 30–34 years with 27.9%, the age group 20–24 years with 19.3%, and then the age group 35–39 years with 15.0%. The admission rate for patients aged age 10–14 years fell by 45.8% (from 0.55 (95% CI: 0.17–0.93) in 2012 to 0.30 (CI 0.04–0.56) in 2020 per 100,000 persons). The admission rate for patients aged 15 years fell by 31.1% [from 7.48 (95% CI: 4.42–10.54) in 2012 to 5.15 (95% CI 2.63–7.67) in 2020 per 100,000 persons]. The admission rate for patients aged 16 years fell by 44.3% (from 24.06 (95% CI 18.65–29.47) in 2012 to 13.41 (95% CI: 9.31–17.52) in 2020 per 100,000 persons). The admission rate for patients aged 17 years fell by 44.6% (from 52.80 (95% CI: 44.77–60.83) in 2012 to 29.24 (95% CI: 23.10–35.38) in 2020 per 100,000 persons). The admission rate for patients aged 18 years fell by 47.2% (from 95.37 (95% CI: 84.62–106.13) in 2012 to 50.37 (95% CI: 42.25–58.48) in 2020 per 100,000 persons). The admission rate for patients aged 19 years fell by 49.5% (from 137.39 (95% CI: 124.75–150.04) in 2012 to 69.44 (95% CI: 60.11–78.76) in 2020 per 100,000 persons). The admission rate for patients aged 20–24 years fell by 35.9% (from 165.47 (95% CI: 159.49–171.45) in 2012 to 106.14 (95% CI: 101.22–111.06) in 2020 per 100,000 persons). The admission rate for patients aged 25–29 years fell by 23.9% (from 210.33 (95% CI: 203.72–216.94) in 2012 to 159.98 (95% CI: 154.22–165.75) in 2020 per 100,000 persons). The admission rate for aged 30–34 years fell by 23.5% (from 213.16 (95% CI: 206.51–219.80) in 2012 to 163.03 (95% CI: 157.31–168.75) in 2020 per 100,000 persons). The admission rate for patients aged 35–39 years fell by 20.1% (from 113.76 (95% CI: 108.68–118.85) in 2012 to 90.86 (95% CI: 86.56–95.16) in 2020 per 100,000 persons). The admission rate for patients aged 40–44 years fell by 5.4% (from 24.53 (95% CI: 22.31–26.74) in 2012 to 23.20 (95% CI: 20.94–25.45) in 2020 per 100,000 persons). The admission rate for patients aged 45–49 years fell by 60.7% (from 1.66 (95% CI: 1.09–2.23) in 2012 to 0.65 (95% CI: 0.28–1.02) in 2020 per 100,000 persons) (Figure 2).

### 3.1. Preterm Labor and Delivery Admission Rate by Age Group

All types of preterm labor and delivery-related hospital admissions were more frequent among the age group from 19 to 39 years (Figure 3).

### 3.2. Other Preterm Labor and Delivery-Related Admission Rates

Other admission method rates for preterm labor and delivery fell by 26.5% (from 99.17 (95% CI: 97.52–100.81) in 2012 to 72.85 (95% CI: 71.44–74.26) in 2020 per 100,000 persons, *p* ≤ 0.05). Emergency admission rates for preterm labor and delivery increased by 39.8% (from 0.80 (95% CI: 0.65–0.95) in 2012 to 1.12 (95% CI: 0.95–1.30) in 2020 per 100,000 persons, *p* ≤ 0.05). Finished consultant episodes’ bed days’ rates for preterm labor and delivery fell by 25.9% (from 360.28 (95% CI: 357.16–363.41) in 2012 to 266.87 (95% CI: 264.18–269.57) in 2020 per 100,000 persons, *p* ≤ 0.05). Waiting list rates for preterm labor and delivery fell by 3.1% (from 0.23 (95% CI: 0.15–0.31) in 2012 to 0.22 (95% CI: 0.14–0.30) in 2020 per 100,000 persons, *p* ≥ 0.05). Planned admission rates for preterm labor and delivery fell by 20.0% (from 0.14 (95% CI: 0.08–0.20) in 2012 to 0.11 (95% CI: 0.06–0.17) in 2020 per 100,000 persons, *p ≤* 0.05) (Figure 4).

## 4. Discussion

The aim of this study was to elucidate the trend of hospital admissions due to preterm labor and birth in England during 2012–2020. The key findings of this study were: (1) The overall hospital admission rates due to preterm pregnancies decreased by 26.2% during the said period. (2) Preterm spontaneous labor with preterm delivery, preterm labor without delivery, and preterm delivery without spontaneous labor were the most common reasons for admissions. (3) The decrease in hospital admissions was due to decreased rates in all the above conditions. (4) Preterm labor and delivery-related hospital admissions were more frequent among the age group from 19 to 39 years.

Recent studies in the UK have demonstrated a huge increase in admissions related to different disease areas across different patient groups [24,25,26]. Insights about the global burden of preterm births are severely limited due to the scarcity of validated data. In low and middle income countries, routine medical surveillance and data collection are poor, while determination of gestational age is challenging. In high income countries, the majority of stillbirths are due to premature delivery [18]. In a report that included data from surveys, national registries, and published and unpublished studies, the estimation of preterm births and its trends over the past twenty years were provided. It showed that the global average of preterm birth rates in 2010 was 11.1%, with the highest occurrences in southern Asia and sub-Saharan Africa. The global contribution of Europe to preterm birth rates was reported to be 5% [27]. In the current study, based on the HES database, the rate of hospital admissions due to preterm conditions was 14,210 per 100,000 persons in 2012, which decreased by 26.2% to about 10,490 per 100,000 persons in 2020.

The etiologies of preterm births can be classified as spontaneous and provider-initiated. In spontaneous preterm births, labor is natural, whereas in the later cases, the patient is deliberately induced to go into labor and deliver (generally due to maternal or fetal indications). Rubens et al. evaluated the relevance of current reproductive science in understanding the causes of preterm births and found that almost 70% of preterm births are spontaneous [28]. In the current study, spontaneous labor with preterm delivery and preterm labor without delivery were found to be the leading causes of hospital admission in England during the study period.

The global community is becoming aware of the increasing rates of preterm births and infant mortality. Thus, in recent times, various interventions have focused on managing the risk associated with it. Quality prenatal care, the maintenance of a record of the patient’s obstetric history, the management of chronic diseases, and maternal counseling are some interventions that have helped in reducing complications and poor outcomes of preterm births [17]. In addition, interventions such as pregnancy spacing, family planning, ensuring optimum nutrition, and maternal support, have helped in reducing preterm birth rates. As a result, the overall hospital admission has witnessed a reduction. Sheu et al. reported a decreased ratio of undelivered to delivered preterm labor during 2007–2014 [29]. This demonstrates that the identification of true preterm labor is significantly improving. Herein, while the total number of hospital admissions decreased by 26.2% during the study period. Simultaneously, the hospital admissions rate due to various etiologies also decreased as follows: preterm delivery without spontaneous labor decreased by 32.0%; preterm labor without delivery decreased by 26.9%; preterm spontaneous labor with preterm delivery decreased by 25.4%; and preterm spontaneous labor with term delivery decreased by 14.7%.

Various factors that interact and cause preterm births include genetics, biology, behavior, and the environment to which the mother is exposed. Furthermore, women with a history of preterm delivery, chronic diseases, uterine pathology, psychological troubles, and demographic risks are more susceptible to preterm labor and birth. In a population-based registry study conducted in Sweden, Waldenström et al. showed that advanced maternal age is a risk factor for preterm birth [28]. However, in this study, 28.4% of total preterm admissions were from women between the ages of 25 and 29, 27.9% were from women between the ages of 30 and 34, and 15% were from women over the age of 35. By 2020, these rates will have dropped by 23.9%, 23.55%, and 20.5%, respectively.

Herein, it was also observed that emergency admission rates for preterm labor and delivery increased by 39.8% between 2012 and 2020. This indicates the increasing burden on hospitalization due to suspicion of preterm labor, which incurs significant socioeconomic costs [29]. Thus, more efforts are necessary to recognize and manage preterm labor and delivery efficiently.

## 5. Conclusions

This study observed that the overall admission rates due to preterm labor and births decreased by 26.2% during the study period. The common etiology of admission was preterm spontaneous labor with preterm delivery, preterm labor without delivery, and preterm delivery without spontaneous labor. A decrease in admission rates due to all the above etiologies was also recorded. All types of preterm labor and delivery-related hospital admissions were more common among the age group from 19 to 39 years, while women aged between 24–39 years were more prone to hospital admission due to preterm labor conditions. Significantly, emergency admission rates for preterm labor and delivery increased by 39.8%. While the decrease in admission rates suggests an improvement in quality health care and a better prognosis for preterm conditions, increased emergency admission rates also indicate an increasing burden on hospitalization. While taking into account the ensuing socio-economic and psychological burden of preterm labor and birth, it is suggested that more attention should be paid to the area for its better management and prevention.

## Figures and Tables

**Figure 1 healthcare-11-00163-f001:**
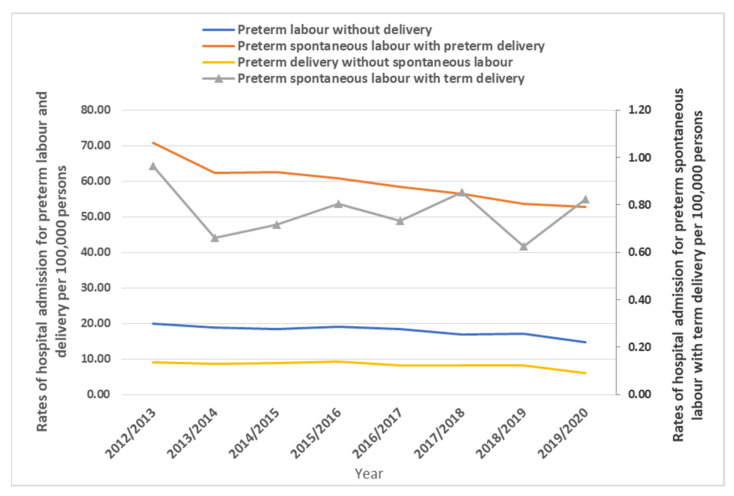
Hospital admission rates stratified by type between 2012 and 2020.

**Figure 2 healthcare-11-00163-f002:**
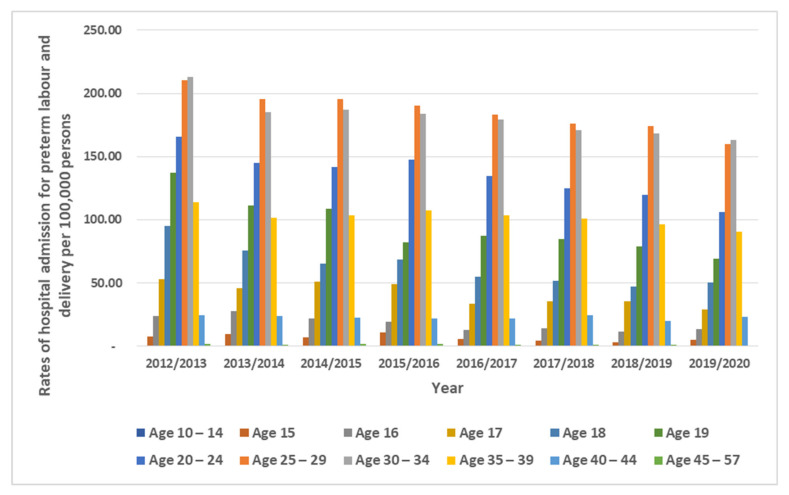
Rates of hospital admission stratified by age group.

**Figure 3 healthcare-11-00163-f003:**
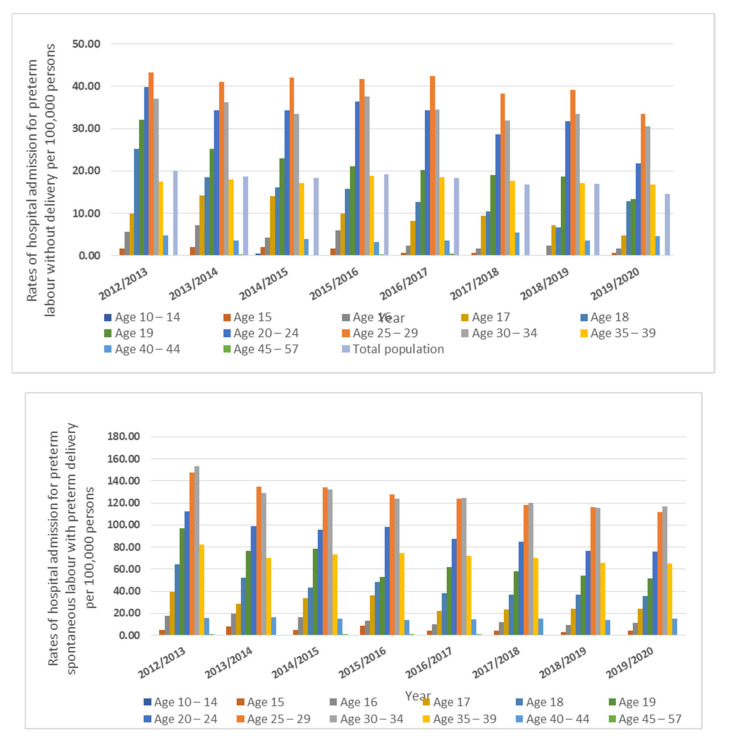
Hospital admission rates stratified by age group.

**Figure 4 healthcare-11-00163-f004:**
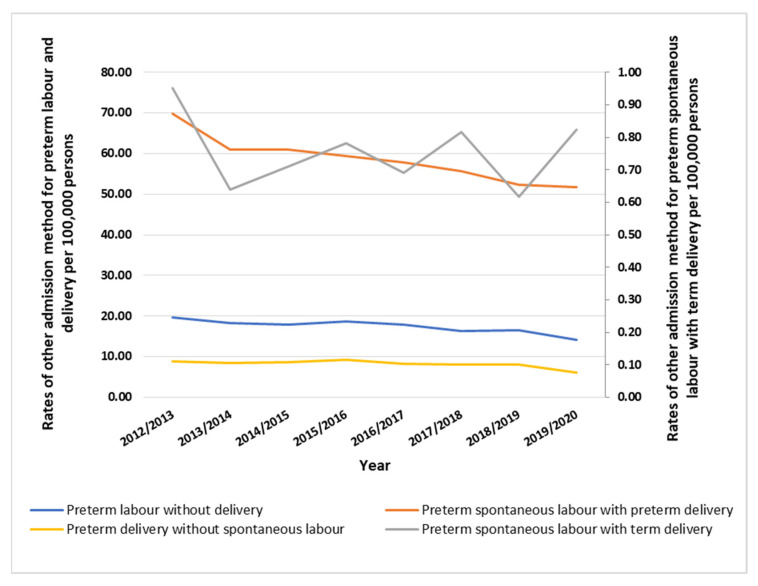
Other preterm labor and delivery-related admission rates in England.

**Table 1 healthcare-11-00163-t001:** Percentage of preterm labor and delivery hospital admission.

ICD Code.	Indication	Percentage
**O60.0**	“Preterm labor without delivery”	20.6%
**O60.1**	“Preterm spontaneous labor with preterm delivery”	68.9%
**O60.2**	“Preterm spontaneous labor with term delivery”	0.9%
**O60.3**	“Preterm delivery without spontaneous labor”	9.6%

ICD International Statistical Classification of Diseases system.

## Data Availability

Publicly available datasets were analyzed in this study. This data can be found here: http://content.digital.nhs.uk/hes and http://www.infoandstats.wales.nhs.uk/page.cfm?pid=41010&orgid=869 (accessed on 23 March 2022).

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
