# Peer review of "Profile of Hospital Admissions Due to Preterm Labor and Delivery in England"

_healthcare, 2023, doi:10.3390/healthcare11020163_

Round 1

Reviewer 1 Report

Dear authors,

Thank you for the paper submitted covering an interesting topic. I think the article tries to  analyze preterm labor and birth-related hospital admissions trend in England. I, however, have major comments and edits that I would like to suggest to the authors: 

- Self-citations: The first author (Naser) has 15 self-citations. I consider this amount of citations unnecessary, and its presence is not justified at all. It seems like this author has written this paper for his own self-citation. Furthermore, 15 of 38 articles in the references belong to the first author Naser (almost half of them).

- Plagiarism: After using an anti-plagiarism tool, I have found that there are literally copied paragraphs from various articles. The one with the highest similarity:

-6% from the article: 

Sweiss K, Naser AY, Alrawashdeh HM, Alharazneh A. Hospital admissions due to vasomotor and allergic rhinitis in England and Wales between 1999 and 2019: an ecological study [published online ahead of print, 2022 Apr 7]. Ir J Med Sci. 2022;10.1007/s11845-022-02996-x. doi:10.1007/s11845-022-02996-x

-References: There is an important issue about the references. There is a duplicity in the Reference section. This circumstance is inexcusable in a submission to a peer-reviewed journal. This circumstance denotes a lack of seriousness in scientific writing. 

Author Response

Thank you for the paper submitted covering an interesting topic. I think the article tries to  analyze preterm labor and birth-related hospital admissions trend in England. I, however, have major comments and edits that I would like to suggest to the authors: 

First of all, we would like to thank the reviewer for the time and efforts in reviewing our manuscript.

- Self-citations: The first author (Naser) has 15 self-citations. I consider this amount of citations unnecessary, and its presence is not justified at all. It seems like this author has written this paper for his own self-citation. Furthermore, 15 of 38 articles in the references belong to the first author Naser (almost half of them).

- Thank you for this comment, the reasons behind citing the authors’ own studies is that Dr. Naser has published multiple studies that explored admissions related to different diseases using this medical database starting from 2017 until the time of the submission of this manuscript. Dr. Naser’s previous publication was cited to highlight the gab in the literature related to preterm labor and birth-related hospital admissions trend in England. All previous studies in the UK focused on specific health outcome or disease with no previous study that examined all admissions related to preterm labor and birth-related to different complications. However, based on the reviewer comment we have no decreased the number of articles cited to highlight the gab in the literature to avoid any confusion.

- Plagiarism: After using an anti-plagiarism tool, I have found that there are literally copied paragraphs from various articles. The one with the highest similarity:

-6% from the article: Sweiss K, Naser AY, Alrawashdeh HM, Alharazneh A. Hospital admissions due to vasomotor and allergic rhinitis in England and Wales between 1999 and 2019: an ecological study [published online ahead of print, 2022 Apr 7]. Ir J Med Sci. 2022;10.1007/s11845-022-02996-x. doi:10.1007/s11845-022-02996-x

- Thank you for this comment, we totally understand the point mentioned by the reviewer in relation to plagiarism that might be faced due to the fact that we are using the same research methodology and same database which necessitate the use of similar procedure of method and results drafting. However, based on the reviewer comment, we have no re-wrote paragraphs with similarities.

-References: There is an important issue about the references. There is a duplicity in the Reference section. This circumstance is inexcusable in a submission to a peer-reviewed journal. This circumstance denotes a lack of seriousness in scientific writing. 

- We highly appreciate the reviewer comments and feedback and totally understand that the goal from them is to enhance the presentation of the manuscript for the readers. Therefore, we have now decreased the number of references and kept the one more relevant to highlight the gab in the literature.

Reviewer 2 Report

There are a couple of things in the article, need to be fixed by the authors.

Author Response

Dear reviewer

Thank you for your time and effort in reviewing our manuscript. 

We have now checked the manuscript in term of grammar mistakes based on your comment. In addition, we have now added the references for the two highlighted statements.

- Regarding your last comment about the impact of COVID-19 pandemic on admission rates for pregnancy and preterm labor, women who contract COVID-19 during pregnancy are at a larger risk of having a very preterm birth, as well as any preterm delivery, given that the prevalence of COVID-19 is higher in these populations. However, we did not add this information as our study did not include data during the pandemic and therefore, we think that our admission rates estimate was not affected by the pandemic.

Reviewer 3 Report

This manuscript is well written, results are significant, statistical analysis is performed in good quality. Although I have only one ethical concern that I think should be explained in method section. The data is from England but all authors and their affiliations are outside from UK. The author should explain if they received any permission to access this data in method section.

Author Response

First of all, we would like to thank the reviewer for the time and efforts in reviewing our manuscript.

Regarding the reviewer comment on the use of UK data by authors from outside the UK, we have known highlighted in the method section that the HES database makes hospital admissions data freely available to the public with no permission needed. Therefore, all researchers could use these data and share the outcome of their study findings to all readers and enhance the provision of healthcare.

Round 2

Reviewer 1 Report

Dear authors:

The paper now reads much better and I believe that it will have a significant contribution to improving knowledge about admissions due to preterm delivery. I, however, have minor comments  that I would like to suggest to the authors.

Abstract :If the word count permits, please structure your abstract with a total of about 200 words maximum.

The number of self-citations have now been decreased, but I  still think there are too many of them. Although the author Naser has published about this regard, as a sign of ethics in scientific publications, the number of self-citations should be less than 6.

The rest of the publication has been properly modified.

Author Response

Thank you for your valuable comments. We have now addressed all comments raised by the reviewer.